# Prevalence of insufficient weight loss 5 years after Roux-en-Y gastric bypass: metabolic consequences and prediction estimates: a prospective registry study

Markus Brissman [1,2] Andrew J Beamish,[3,4] Torsten Olbers,[5,6] Claude Marcus[1]

► Prepublication history and supplemental material for this paper is available online. To view these files, please visit the journal online (http://dx.doi.org/10.1136/bmjopen-2020-046407).

For numbered affiliations see end of article.

**Correspondence to**
Dr Markus Brissman;
markus.brissman@ki.se

## ABSTRACT

**Objective** The study aimed to investigate the heterogeneity of weight loss 5 years after Roux-en-Y gastric bypass (RYGB) and the association with cardiometabolic health as well as to model prediction estimates of surgical treatment failure.

**Design** Retrospective analysis of prospectively collected data from the Scandinavian Obesity Surgery Registry (SOReg).

**Setting** 29 surgical units from the whole of Sweden contributed data. Inclusion was restricted to surgical units with a retention rate of >60% five years postsurgery.

**Participants** 10 633 patients were extracted from SOReg. In total 5936 participants were included in the final sample, 79.1% females. The mean age of participants before surgery was 39.4±9 years and mean body mass index (BMI) 42.9±5.1. 2322 were excluded (death before the 5-year follow-up (n=148), other types of surgery or reoperations (n=637), age at surgery <18 or >55 years (n=1329), presurgery BMI <35 kg/m² (n=208)). In total, 2375 (29%) of eligible individuals were lost to the 5-year follow-up.

**Main outcome** The occurrence of surgical treatment failure 5 years after surgery was based on the three previously published definitions: per cent excess BMI loss <50%, total weight loss <20% or BMI >35 where initial BMI was <50, or >40 where initial BMI was >50. In addition, we report the association between surgical treatment failure and biochemical markers of obesity-related comorbidity. We also developed predictive models to identify patients with a high risk of surgical treatment failure 5 years postsurgery.

**Results** In total, 23.1% met at least one definition of surgical treatment failure at year 5 which was associated with (adjusted OR) with 95% CI): type 2 diabetes (T2D, OR 2.1; 95% CI 1.6 to 2.7), dyslipidaemia (OR 1.8; 95% CI 1.6 to 2.1) and hypertension (OR 1.9; 95% CI 1.6 to 2.2). Surgical treatment failure at 5 years was predicted by combined demographic and anthropometric measures from baseline, 1 and 2 years postsurgery (area under the curve=0.874).

**Conclusion** Laparoscopic RYGB leads to a marked and sustained weight loss with improvement of obesity-related comorbidity in most patients. However, 23% met at least one definition of surgical treatment failure, which was associated with a greater risk of relapse and a higher incidence of T2D, dyslipidaemia and hypertension 5 years

after surgery. Poor initial weight loss and early weight regain are strong predictors of long-term treatment failure and may be used for early identification of patients who require additional weight loss support.

## Strengths and limitations of this study

► A large prospective cohort of nearly 6000 patients from bariatric surgery centres with a minimum of 60% retention rate at year 5 after bariatric surgery.
► Predefined thresholds of surgical treatment failure and cardiometabolic health were applied.
► The prediction model of surgical treatment failure was cross-validated using partial data, however, further validation of an unrelated cohort is preferable.
► Data originate from the whole of Sweden; thus generalisability may be limited to countries with similar ethnic diversity.

## INTRODUCTION

Obesity is a heterogeneous disease[1] associated with several comorbid conditions, which ultimately increases the risk of all-cause mortality.[2] Bariatric surgery is the most effective treatment for severe obesity. Long-term follow-up studies of Roux-en-Y gastric bypass (RYGB) show excellent results at the group level in reductions in weight, morbidity and mortality compared with non-surgical treatment.[3–5] In Sweden, approximately 5500 bariatric operations are performed annually and, until 2014, the technique was almost exclusively RYGB.[6]

Weight loss after surgery is typically achieved during the first and second year, followed by weight maintenance or moderate regain 5–10 years after surgery.[7] However, despite good overall results, the response and durability of surgically induced weight loss are heterogeneous[8–10] and surgical treatment failure has been recognised as a potential clinical problem.[11–13]

The prevalence of surgical treatment failure is unclear, largely because an all-encompassing, unambiguous definition remains elusive.[11–14] In a landmark controlled study by Adams *et al*[5] based on 418 RYGB patients, 30% of participants experienced <20% of total body weight loss at 12 years after RYGB.

It is still unclear to which extent cardiometabolic improvements after bariatric surgery depends on the degree of weight loss. Long-term studies have reported temporally declining rates of remission from obesity-related comorbidities[5 15] and the rate of relapse, especially for type 2 diabetes (T2D), has rather been attributed to pre-surgery disease duration and progression than to insufficient weight loss.[5 16] Although an association between T2D relapse and weight regain has been suggested in some studies,[17–19] others have not found any association between the degree of long-term weight loss and cardiometabolic outcome.[20–22] The annual summary of the Scandinavian Obesity Surgery Registry (SOReg) recently described an association between baseline T2D and inadequate postoperative weight loss.[23]

In this study, based on a large cohort of patients prospectively collected in SOReg,[6] we report on the heterogeneity of weight loss outcome, focusing primarily on the occurrence of surgical treatment failure 5 years after surgery, according to any of three published definitions. We also report the association between surgical treatment failure and cardiometabolic disease and we present predictions of surgical treatment failure based on background data and weight development during the first 2 years after RYGB.

## METHODS
### Data source
The data source for this study was SOReg, a Swedish nationwide registry that began collecting data in 2007; from 2011, the registry covered 95%–99% of all bariatric surgery performed in Sweden. Between 2007 and 2011, RYGB constituted 96%–97% of all bariatric surgery performed. Data were retrieved in accordance with the study protocol. For this retrospective analysis, data were requested for all patients from surgical units and yearly cohorts that had a 5-year retention rate of ≥60%. Data covered demographics, anthropometrics, pharmacological treatment, obesity-related comorbidity, biochemical markers and blood pressure at four time points: before surgery (baseline), and at 1, 2 and 5 years after surgery.

### Participants
In total, 29 surgical units contributed data to the study through the SOReg database, ranging in number from 1 to 1643 patients, and data on 10 633 unique patients were extracted.

Exclusion was performed in iteration steps and a total of 4697 patients were excluded.

The participants included in this study underwent RYGB during 2007–2011, 84.3% had body mass index (BMI) reported for all time points. Missing data on BMI totalled 13.2% at either the 1-year or 2-year follow-up,

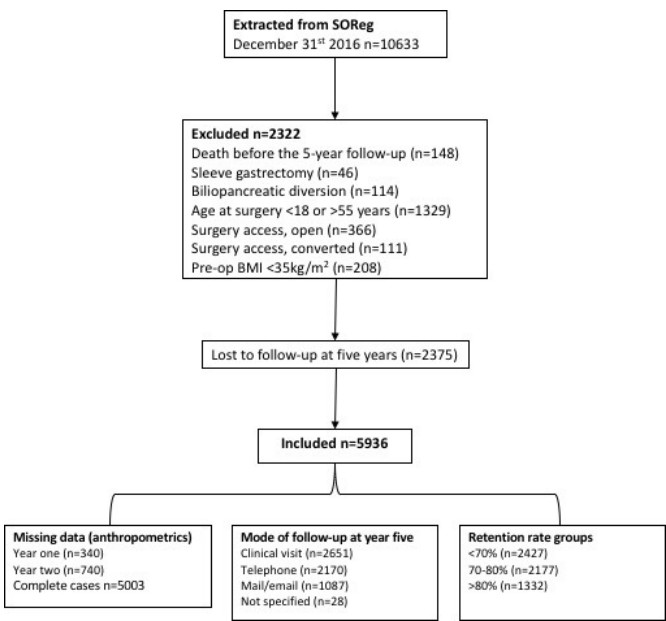

**Figure 1** Flow chart of the study participants. BMI, body mass index; SOReg, Scandinavian Obesity Surgery Registry.

and 2.5% at both the 1-year and 2-year follow-ups. The follow-up modality at the 5-year follow-up was a clinical visit (44.7%), telephone consultation (36.6%), email/letter (18.3%) or unspecified (0.5%). The follow-up modality at the 1-year and 2-year follow-ups are presented in detail in online supplemental eTable 1. Figure 1 shows the flow chart of the study participants.

### Loss to follow-up analysis
A comparison of baseline characteristics between the study participants and those lost to follow-up revealed that lost participants had a younger age, a higher BMI and a male predominance. A detailed comparison appears in online supplemental eTable 2 in the Supplement.

## DEFINITIONS
### Surgical treatment failure
Surgical treatment failure was assessed and defined as meeting at least one of three definitions 5 years after surgery: (1) <50% excess BMI loss (%EBMIL), (2) <20% total weight loss (%TWL) and (3) BMI >35 kg/m$^2$ where baseline was <50 kg/m$^2$, or >40 kg/m$^2$ if baseline BMI was >50 kg/m$^2$. These definitions have been used elsewhere[11 24] and, taken together, provide a means to define failure for patients within different weight categories.

%EBMIL was calculated as ((baseline BMI − year 5 BMI)/(baseline BMI − 25))*100.

%TWL was calculated as ((baseline BMI − year 5 BMI)/baseline BMI)*100.

Two trajectories—inadequate weight loss and weight regain—can be defined that lead to long-term surgical treatment failure. Inadequate weight loss has been quantified during the first 6–12 months after surgery,[25] and weight regain has typically been described as an increase above a specified threshold.[12 13]

In this paper, inadequate weight loss was defined as <25% TWL from baseline to 1-year postsurgery, similar to the 25th percentile presented by Manning et al.[25]

Early weight regain was defined as any absolute weight gain, expressed in kilograms, occurring between year 1 and 2 after surgery. This definition generated two groups. Long-term weight regain, defined according to Odom et al[26] in three groups: >15% regain of BMI nadir, 0.1%–15% regain of BMI nadir and no weight regain, to 5 years postsurgery. These definitions were used to capture early weight regain as a predictive measurement of long-term surgical treatment failure, and to differentiate between the normally occurring fluctuation of body weight in the maintenance phase and the potentially harmful weight regain previously suggested.[18 19]

For calculations, BMI nadir was accepted as the lowest measured weight at either the 1 or 2-year follow-up. In the case of missing data from one of those time points, the observed measurement was taken as the nadir.

## Obesity-related comorbidities and metabolic markers

It is mandatory to report obesity-related comorbidities (eg, T2D, dyslipidaemia, hypertension)[6] requiring pharmacological treatment in SOReg, and data were available for 88%–100% included individuals depending on time point (full description in online supplemental eTable 3).

Blood pressure and biochemical markers, such as low-density lipoprotein, high-density lipoprotein, triglycerides, fasting glucose and glycated haemoglobin, are optional to report. Data were available data from 34%–73% included participants (online supplemental eTable 3).

Changes in blood pressure and biochemical markers were compared, stratified by surgical treatment failure at the 5-year follow-up, and by pharmacological treatment at baseline. Additionally, a broader classification of disease traits was generated, similar to that previously described,[5 27] by compiling a disease-specific biochemical marker above a cut-off (online supplemental eAppendix 1), in combination with pharmacological treatment. This classification was applied at all time points and used to assess prevalence and change over time. Thus, six groups were generated: participants without disease traits at baseline were classified 'disease-free' if no disease trait was evident at any time point, 'intermittent' if disease-free at both baseline and 5-year follow-up, but not in between, and 'incidence' where a disease trait developed during the 5-year follow-up period. Participants with a disease trait at baseline were classified 'remission' if no disease trait was evident at 5-year follow-up, 'relapse' if disease-free at year 1, 2 or both, but not at year 5, and 'no remission' where at least one disease trait was evident at all time points.

For clarity, the compiled disease traits are hereafter referred to as T2D, dyslipidaemia and hypertension.

## Statistics

All statistical analyses were performed using SPSS V.24 (IBM) and STATA IC V.15.1 (Stata). Descriptive statistics are presented as mean±standard deviation (±SD), or as a percentage (%), unless otherwise specified.

Characteristics were compared between those lost to follow-up (online supplemental eTable 2) and those included in the analysis, as well as according to surgical treatment failure status (online supplemental eTable 4), using independent t-test and $\chi^2$ test.

We described the prevalence and change in cardiometabolic disease and assessed the odds associated with surgical treatment failure using logistic regression, first using a crude model (data not shown) and then multivariable models (separate, compiled or additive for each definition of surgical treatment failure) in which we adjusted for sex, age and BMI at baseline and corresponding cardiometabolic disease. Results are presented as OR with 95% CIs.

In addition, we used logistic regression to predict the probability of meeting at least one definition of surgical treatment failure, which we considered dichotomously (1=surgical treatment failure, 0=otherwise). Our predictions used sex, baseline, age, BMI and %TWL for the first year and change in weight (kg) for the second year. We measured performance by calculating the receiver operating characteristic curve and the corresponding area under the curve (AUC) and by using cross-validation (leave 10%, k=10 replicates).

Finally, several sensitivity analyses were undertaken for the primary endpoint (ie, surgical treatment failure), which can be found in online supplemental eAppendix 2.

The significance level was set to 0.05 for all analyses (two tailed), and p values are reported with three decimals.

## Patient and public involvement

Patients nor the public were involved in the conduct of this study.

## RESULTS

In total, 5936 patients (79.1% female), aged 18–55 years, who had undergone laparoscopic RYGB (LRYGB) from 2007 to 2011, were included in the final sample (figure 1). At baseline, the mean age was 39.4±9.0 years and BMI was 42.9±5.1 kg/m². Patient characteristics are presented in table 1. At year 5, overall mean BMI was 30.4±5.2, mean weight loss 35.8±13.8 kg, BMIL 12.6±4.7 kg/m2, %EBMIL 72.2±25.2% and %TWL 29.1±9.8%.

Inadequate weight loss (ie, <25% TWL from baseline to year 1) was identified in 17.1% of 5596 participants with available data.

Early weight regain (between year 1 and 2) was identified in 38.7% of 5010 participants with available data, with a mean increase of 4.5±3.9 kg (range 1–38 kg), compared with a mean decrease of 4.4±5.1 kg (range 66–0 kg) in the no regain group.

Long-term weight change between nadir and 5-year follow-up was distributed as follows:>15% regain (+17.7±7.2 kg, range 7–101 kg) in 19.9% of participants, 0.1%–15% regain (+5.7±3.5 kg, range 0 to 19 kg) in 59.3%

**Table 1** Baseline characteristics of the study population

| | n | Mean (SD) |
|---|---|---|
| Age at surgery | 5936 | 39.4 (9.0) |
| Sex, no % female | 5936 | 79.1 |
| Height, cm | 5936 | 168.8 (8.9) |
| Weight, kg | 5936 | 122.8 (20.0) |
| Body mass index at surgery, kg/m* | 5936 | 42.9 (5.1) |
| Glucose metabolism | | |
| Glucose, mmol/L | 2861 | 5.9 (1.9) |
| HbA1c, mmol/mol | 4168 | 40.6 (11.4) |
| Pharmacological diabetes treatment, no (%) | 5936 | 675 (11.4) |
| Diabetes type 2†, no (%) | 5936 | 896 15.1 |
| Lipids | | |
| High-density lipoprotein (HDL), mmol/L | 4188 | 1.2 (0.4) |
| Low-density lipoprotein (LDL), mmol/L | 4110 | 3.1 (0.9) |
| Triglycerides (TG), mmol/L | 4314 | 1.7 (1.4) |
| Pharmacological dyslipidaemia treatment, no (%) | 5936 | 414 (7.0) |
| Dyslipidaemia‡, no (%) | 5936 | 3601 (67.5) |
| Physiology | | |
| Systolic BP, mm Hg | 2960 | 133 (16) |
| Diastolic BP, mm Hg | 2960 | 83 (10) |
| Pharmacological hypertension treatment, no (%) | 5936 | 1158 (19.5) |
| Hypertension‡ no. (%) | 5936 | 1683 (28.4) |

*Pharmacologically treated T2D/fasting glucose >7.0 mmol/L/ HbA1c >48 mmol/mol.
†Pharmacologically treated dyslipidaemia/LDL >4.1/TG >2.0/HDL <1 mmol/L for males and <1.3 mmol/L for females.
‡Pharmacologically treated blood pressure/systolic≥140 mm Hg or diastolic blood pressure >90 mm Hg.
BP, blood pressure; HbA1c, glycated haemoglobin; T2D, type 2 diabetes.

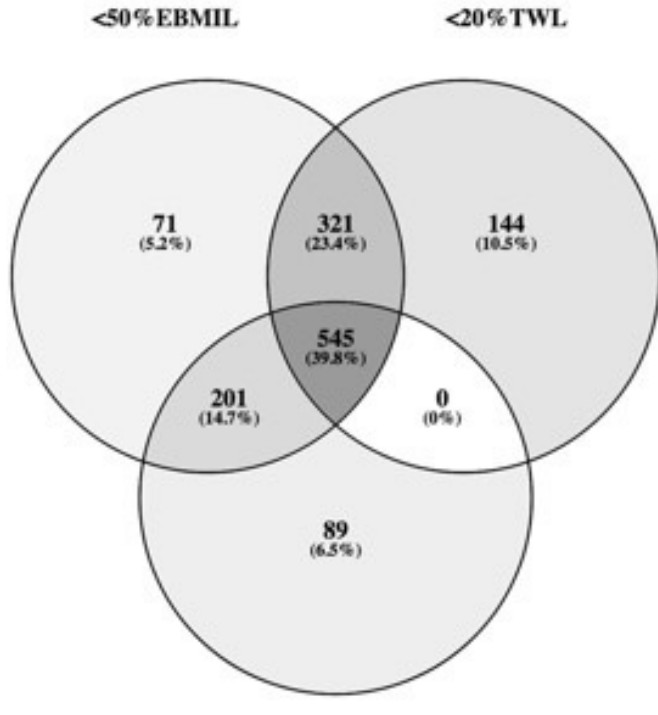

**Figure 2** Venn diagram of the prevalence of developing surgical treatment failure 5 years postsurgery according to three definitions: %excess BMI loss (n=1138), BMI >35 or >40 (n=835) and <20% total weight loss (n=1010). BMI, body mass index; EBMIL; excess BMI loss; TWL, total weight loss.

and no weight regain (−5.0±5.2 kg, range −36 to 0 kg) in 20.8%.

Overall, the prevalence of meeting at least one of the three definitions of surgical treatment failure 5 years after LRYGB was 23.1% (n=1371). The distribution between the three definitions was 19.2% (n=1138) for <50%EBMIL, 17.0% (n=1010) for <20% TWL and 14.1% (n=835) for BMI>35 or>40 kg/m². There was substantial overlap, 39.8% (n=545) meeting all three definitions and 38.1% (n=522) meeting two of the three definitions (figure 2).

Surgical treatment failure was more common among patients with inadequate weight loss (60% vs 15.4%, p<.001) and early weight regain (33.8% vs 15.6%, p<.001). Comparing long-term weight regain, the proportion meeting criteria for failure was highest in participants with >15% regain from nadir (46.7%), followed by 0.1%–15% (21.1%) and no regain (5.1%), (p<.001). Patients with no long-term weight regain but surgical treatment failure (n=59) had higher baseline BMI (48.5 vs 43.1, p<.001) and lower %TWL at 1-year and 2-year follow-up (−18.0% vs −30.5% and −18.1% vs −32.3%, respectively, both p<.001).

### Cardiometabolic disease

Biochemical and physiological measures improved following surgery in participants with and without surgical treatment failure. Mean values, stratified by surgical treatment failure and baseline pharmacological treatment, are shown from baseline to year 5 in online supplemental eFigures 1A–G and 2A–G.

Overall, the prevalence of cardiometabolic disease decreased from baseline to 5 years: T2D from 15.1% (n=896) to 6.4% (n=380), dyslipidaemia from 60.7% (n=3603) to 16.4% (n=974), and hypertension from 28.4% (n=1683) to 18.9% (n=1124). The rates of being disease-free, incident and intermittent disease, as well as remission, relapse and no remission, varied between surgical and non-surgical treatment failure (table 2).

Logistic regression (adjusted for sex, age, BMI and corresponding cardiometabolic disease at baseline) confirmed an association between surgical treatment failure and cardiometabolic disease at year 5: T2D,

**Table 2** Change in cardiometabolic disease status from baseline to 5 years postsurgery compared between surgical treatment failure (STF) and non-STF

| | Type 2 diabetes (T2D)* | | Dyslipidaemia† | | Hypertension‡ | |
|---|---|---|---|---|---|---|
| | STF n=1135 | Non-STF n=3878 | STF n=1120 | Non-STF n=3867 | STF n=1126 | Non-STF n=3842 |
| No disease at baseline | n=882, % | n=3379, % | n=377, % | n=1616, % | n=735, % | n=2818, % |
| Disease-free | 97.4 | 98.5 | 82.0 | 87.3 | 83.9§ | 91.3 |
| Incidence | 1.6¶ | 0.7 | 9.5§ | 4.9 | 9.9§ | 4.6 |
| Intermittent | 1.0 | 0.9 | 8.5 | 7.8 | 6.1‡ | 4.0 |
| Disease at baseline | n=253, % | n=499, % | n=743, % | n=2251, % | n=391, % | n=1024, % |
| Remission | 51.4§ | 66.5 | 63.7§ | 81.1 | 38.6§ | 54.6 |
| No remission | 26.1 | 22.4 | 17.2§ | 8.8 | 37.6§ | 27.1 |
| Relapse | 22.5§ | 11.0 | 19.1§ | 10.1 | 23.8** | 18.3 |

*Pharmacologically treated T2D/fasting glucose >7.0 mmol/L/HbA1c>48 mmol/mol.
†Pharmacologically treated dyslipidaemia/LDL >4.1/TG >2.0/HDL <1 mmol/L for males and <1.3 mmol/L for females.
‡Pharmacologically treated blood pressure/systolic ≥140 mm Hg or diastolic blood pressure >90 mm Hg.
§Indicates a statistically significant difference at p<.001.
¶Indicates a statistically significant difference at p<.010.
**Indicates a statistically significant difference at p<.001.
HDL, high-density lipoprotein; LDL, low-density lipoprotein; T2D, type 2 diabetes; TG, triglycerides.

OR 2.10 (95% CI 1.61 to 2.75); dyslipidaemia, OR 2.50 (95% CI 2.14 to 2.92) and hypertension, OR 1.85 (95% CI 1.55 to 2.21). Individual definitions were similarly associated with cardiometabolic disease (online supplemental eTable 5). The combined effect of fulfilling one, two or three of the definitions is presented in online supplemental eTable 6. Predicted probability of cardiometabolic disease plotted against continuous %EBMIL, %TWL and BMI at year 5 is illustrated in online supplemental eFigures 3–5A–C.

Inadequate weight loss during year 1 was significantly associated with T2D (OR 1.84; 95% CI 1.38 to 2.45), dyslipidaemia (OR 1.89; 95% CI 1.59 to 2.25) and hypertension (OR 1.61; 95% CI 1.32 to 1.96). Late weight regain (≥15% regain from nadir) was significantly associated with dyslipidaemia (OR 1.64; 95% CI 1.31 to 2.05) and hypertension (OR 1.41; 95% CI 1.10 to 1.81), but not T2D (OR 1.25; 95% CI 0.84 to 1.88).

### Predicting surgical treatment failure

The estimated regression coefficients and OR are presented in table 3. Given age, sex and baseline BMI and a negatively expressed value of %TWL from baseline to the 1-year follow-up, and change in weight (kg) between 1-year and 2-year follow-up, the predicted probability of surgical treatment failure 5 years after surgery is given by:

P(surgical treatment failure)=exp(a)/(1+(a)) with a = −1.1 + −0.00545*(sex male=0 female=1)+0.00299*(age at surgery, years)+0.14949*(baseline BMI)+0.22310*(%TWL year 1)+0.15982*(weight change year 1 to year 2 (kg)). Examples of the probability calculation are presented in online supplemental eAppendix 3.

As depicted in figure 3, this simple model provided a good prediction (AUC=0.8743).

### DISCUSSION

This analysis of prospectively collected data on 5963 adults who underwent primary LRYGB surgery, revealed that almost one in four participants fulfilled at least one of the three applied definitions of surgical treatment failure, 5 years after surgery. Surgical treatment failure was associated with a negative effect on cardiometabolic health: lower rate of remission and more frequent relapse and incidence of T2D, dyslipidaemia and hypertension.

**Table 3** Final multivariable model for predicting surgical treatment failure 5 years after surgery

| | Beta (B) | SE | Wald | P value | Exp(B) | 95% CI |
|---|---|---|---|---|---|---|
| Sex (0=male) | −0.00545 | 0.099 | 0.003 | 0.956 | 0.995 | 0.818 to 1.209 |
| Age at surgery, years | 0.00299 | 0.005 | 0.361 | 0.548 | 1.003 | 0.993 to 1.013 |
| BMI at surgery, kg/m² | 0.14949 | 0.009 | 283.640 | 0.000 | 1.161 | 1.141 to 1.182 |
| Percentage BMI loss during year 1, %TWL | 0.22310 | 0.008 | 794.848 | 0.000 | 1.250 | 1.231 to 1.269 |
| Change in weight between year 1 and 2, kg | 0.15982 | 0.008 | 382.606 | 0.000 | 1.173 | 1.155 to 1.192 |
| Intercept | −1.09588 | 0.513 | 4.569 | 0.033 | 0.334 | |

BMI, body mass index; TWL, total weight loss.

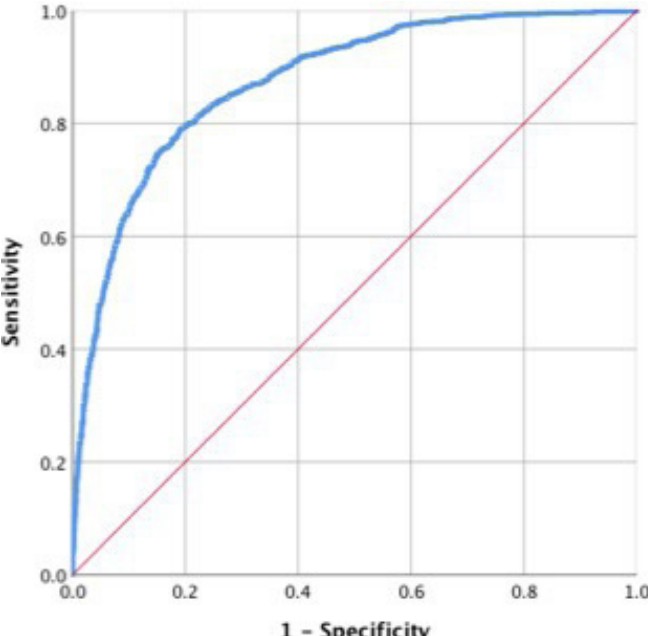

**Figure 3** Receiver operating characteristic curve with predicted probability of surgical treatment failure, given age, sex and baseline BMI and %TWL from baseline to the 1-year follow-up and change in weight (kg) between year 1 and year 2 follow-ups: area under the curve=0.8743 (95% CI 0.8630–0.8856). BMI, body mass index; TWL, total weight loss.

Each definition of surgical treatment failure and weight regain was independently associated with cardiometabolic health. In total, 9.2% of the study population fulfilled the criteria for all of the three definitions[11 24] of surgical treatment failure and they provided a very strong association with T2D, dyslipidaemia and hypertension.

The extent to which insufficient weight loss and weight regain affect cardiometabolic outcome is unclear, both confirmative[13 18 19 28–30] and negative[20–22 31 32] findings have been reported. In the present study inadequate weight loss during year 1 and weight regain during year 2 were investigated. Both were found to be associated with cardiometabolic outcomes, however, both were in the present study viewed as prerequisites for surgical treatment failure, which in turn was associated with a less favourable metabolic profile 5 years after surgery, regardless of whether or not patients were taking T2D, dyslipidaemia or hypertension medications prior to surgery.

Early identification of those with a high risk of long-term surgical treatment failure may facilitate additional weight loss support.[33–35] Unfortunately neither we, nor others, have been able to build a sufficiently reliable model using exclusively presurgical characteristics.[36] However, our results indicate that long-term surgical treatment failure can, with good accuracy (AUC=0.8743), be predicted by sex, age and BMI at baseline, together with %TWL during year 1 and weight change during year 2. We found that %TWL during year 1 was the strongest predictor of surgical treatment failure. Similarly the initial 6-month weight loss predicts the 24-month weight loss.[25] Of note,

we found that presurgical T2D, dyslipidaemia and hypertension were associated with surgical treatment failure, a finding that may warrant further research as the associations could be dependent on both behavioural and physiological factors.

The present study terms long-term poor weight loss after surgery as surgical treatment failure. This wording should not be interpreted to mean that the surgical procedure failed, but rather that the therapy alone was insufficient to produce the required degree of long-term weight loss. This reasoning should not be surprising given the heterogeneous nature of obesity, as any standardised treatment is likely to result in a spectrum of outcomes. Despite that, bariatric surgery has remained a stand-alone treatment. This is contrary to bariatric surgery guidelines suggesting active treatment of patients with poor weight outcome.[37] In addition patients have also expressed a need for more extensive follow-up.[38] Recognising this, bariatric surgery would likely benefit from the application of the multidisciplinary and multimodal approach that has evolved in other fields of disease, such as cancer care, where for decades surgery has been integrated into multimodal treatment pathways, alongside chemotherapies and radiation therapies. It has been shown that behavioural support[35] and pharmacological treatment[34] can improve the outcome after surgery, indicating potential for additive, perhaps even synergistic effects of combination therapies. However, as a consequence of the disintegrated follow-up after surgery, it is still unclear to which extent outcome after bariatric surgery can be optimised by means of adjuvant treatment.

Strengths of this study include SOReg's prospective collection of data from the whole of Sweden, with broad national coverage. This was demonstrated by the inclusion of nearly 6000 patients from the database of centres with a≥60% retention rate 5 years after LRYGB, providing a large and robust data set permitting subgroup analysis. All patients included in the final sample had undergone LRYGB. This constituted 95%–97.5% of all bariatric surgery performed between 2007 and 2011 in Sweden, thus reducing possible bias in patient selection for different surgical procedures.

There are also some limitations. Although the impact of surgical treatment failure on metabolic health is substantial, it does not account for all comorbidity seen at the 5-year follow-up. Other factors, such as disease duration before surgery, are also of importance but such information was not available in this study. Neither was information on psychological disorders available, thus limiting the possibility to evaluate and include such factors in the prediction model.

Missing data analysis revealed that rates of surgical treatment failure at year 1 and 2 were higher in the 28.6% that were lost to follow-up year 5, indicating that the actual proportion of surgical treatment failure may be higher than what the results suggests. In addition, there was a difference in weight loss between the modes of follow-up, possibly implying bias of self-reported data. Similarly, a

statistical limitation of note is that we compiled disease-specific traits where missing data are implicitly treated as zeroes. For example, the estimated effects may be diluted (biased towards zero) because the comparison is actual ones vs a mixture of zeroes and ones. Thus, both the overall prevalence of surgical treatment failure and cardiometabolic disease may be underestimated.

The developed prediction model for long-term surgical treatment failure was cross-validated using partial data and can readily be applied to countries with similar cultural and ethnic settings as in northern Europe. However, further validation of an unrelated cohort is preferable, and further devolvement of the model may be required to encompass ethnic diversity.

Unsuccessful surgical treatment result is difficult to define and a large number of definitions and time points have been used.[11–14] Our results would probably have been slightly modified if we had used other definitions. However, the strong associations between surgical treatment failure, as defined in the present study, and cardiometabolic health may support their clinical usefulness.

## CONCLUSION

RYGB is associated with improvement of obesity-related comorbidity. However, 23% of the patients developed surgical treatment failure 5 years after surgery, which was associated with a markedly increased risk of cardiometabolic disease. Initial weight loss and early weight regain were strong predictive markers that can be used for the early identification of patients with a high risk of long-term failure. This study underlines the need for long-term follow-up of patients undergoing bariatric surgery by a multidisciplinary team and improved additional behavioural and pharmacological treatment postsurgery are warranted.

**Author affiliations**
[1]Department of Clinical Science, Intervention and Technology, Division of Pediatrics, KI CLINTEC, Huddinge, Sweden
[2]Allied Health Professionals Function, Occupational Therapy & Physiotherapy, Karolinska University Hospital, Stockholm, Sweden
[3]Department of Surgical Research, University of Gothenburg Institute of Clinical Sciences, Gothenburg, Sweden
[4]Swansea University Medical School, Swansea, UK
[5]Department of Biomedical and Clinical Sciences, Linkoping University, Linkoping, Sweden
[6]Department of Surgery, Vrinnevi Hospital in Norrkoping, Norrkoping, Sweden

**Contributors** MB and CM conceptualised the study, MB performed data and statistical analyses and drafted the manuscript. All authors (MB, AJB, TO and CM) contributed to result interpretation and critically revised and approved the final version of the manuscript. MB and CM had full access to all the data in the study and takes responsibility for the integrity of the data and the accuracy of the data analysis.

**Funding** This work was funded by the Swedish order of Freemasons (grant no: not applicable), the Swedish Heart- Lung Foundation (grant no: 20180581), the Samariten foundation for pediatric research (grant no: not applicable) and Anna-Lisa and Arne Gustafssons foundation (grant no: not applicable).

**Disclaimer** The funders of this study had no part in study design, collection, analysis or interpretation of data, nor in the writing of the report or in the decision to submit the paper for publication. The corresponding author had full access to the data and had final responsibility for the decision to submit for publication.

**Competing interests** TO declares participation in advisory board for J&J and Novo Nordisk and reimbursement for lectures and education activities. All fees to institution. CM has received research grants from Novo Nordisk, Sigrid THX AB and salaries as medical advisor for Itrim AB and Weight Watchers Int. MB and AJB declares no conflict of interest.

**Patient consent for publication** Not required.

**Ethics approval** This study was approved by the Stockholm ethical board (2017/1793-31).

**Provenance and peer review** Not commissioned; externally peer reviewed.

**Data availability statement** Data may be obtained from a third party and are not publicly available. All data relevant to the study are included in the article or uploaded as online supplemental information. No additional data available from the authors. Original data may be requested from the Scandinavian Obesity Surgery Registry (https://www.ucr.uu.se/soreg/in-english)

**ORCID iD**
Markus Brissman http://orcid.org/0000-0003-1971-6431

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
