## [Reviewer comments · BMJ Open]

ARTICLE DETAILS

TITLE (PROVISIONAL)	Prevalence of insufficient weight loss five years after Roux-en-Y gastric bypass: Prevalence, metabolic consequences and prediction estimates -A prospective registry study
AUTHORS	Brissman, Markus; Beamish, Andrew; Olbers, Torsten; Marcus, Claude

VERSION 1 – REVIEW

REVIEWER	Pontiroli, Antonio University of Milan, medicina
REVIEW RETURNED	24-Dec-2020

GENERAL COMMENTS	Insufficient weight loss five years after Roux-en-Y gastric bypass: Prevalence, metabolic consequences and prediction estimates -A prospective registry study In a retrospective analysis of data prospectively collected from the Scandinavian Obesity Surgery Registry (SOReg), the study aimed to investigate the heterogeneity of weight loss five years after RYGB and the association with cardiometabolic health; in total, 23.1% met at least one definition of surgical treatment failure at five years, and surgical treatment failure at five years was predicted by combined demographic and anthropometric measures from baseline, one and two years post-surgery (area under the curve=0.874). The authors conclude that 23% met at least one criterion of surgical treatment failure, which was associated with a greater risk of relapse and a higher incidence of T2DM, dyslipidemia and hypertension five years after surgery. Poor initial weight loss and early weight regain are strong predictors of long-term treatment failure and may be used for early identification of patients who require additional weight loss support The study is of interest. However, more details should be given for this retrospective study, whenever possible to retrieve. 1. For instance, even though the study is mainly centered on RYGB, we know from the SOS experience that many obese patients underwent other surgical procedures, such as VBG and LAGB. Do the authors have data on these two techniques? Are the SOS and the SOReg completely independent?2. An interesting point of the study is that patients unavailable were different from patients remaining in the follow-up study; do the authors have data on the psycho aspects of these patients, as the work up of patients undergoing bariatric surgery usually includes these aspects? Do they have data on the adherence of patients of the two groups to pre- and post surgery visits and exams, at least for the first two years after surgery?
--

	3. Aside from classical metabolic risk factors, do the authors have data on sleep apnea and its profile over time? 4. Do the authors have data on the predictive role of T2DM and of hypertension for weight loss in general and on the occurrence of STF? Minor points In the text, it should be alluded to male sex and not to sex In the conclusions, the authors should address the unsolved question: is it that bariatric surgery does not respond to expectations in more severe cases? Do the authors have an overall different conclusion?
--	--

REVIEWER	Carlos Aurelio Schiavon Obesity and Metabolic Disease Center - BP Hospital Research Institute - HCor
REVIEW RETURNED	22-Jan-2021

GENERAL COMMENTS	Dear authors, This is a very interesting paper about an important issue. Please, see my comments and try to respond them. Title: I suggest to you to change the order of words: Prevalence of Insufficient weight loss five years after Roux-en-Y gastric bypass: metabolic consequences and prediction estimates - A prospective registry study Objective: I suggest to you to include the prediction estimate in the objectives. Strengths and limitations of this study: You should include in the limitation the number of patients (29%) who lost the FU because this data can interfere in the results because we know that many patients who do not return in the FU have obesity relapse. Participants: Can you present the data (%) about each modality of clinical visit. Page 1, line 13. You should change "incidence" by "de novo" Page 14: $P(\text{surgical treatment failure}) = \exp(a)/(1+(a))$ with $a = -1.1 + 0.00545*(\text{sexFemale}) + 0.00299*(\text{age at surgery}) + 0.14949*(\text{baseline BMI}) + 0.22310*(\%T\text{WL year one}) + 0.15982*(\text{weight change year one to year two (kg)})$. You need to explain this formula. I really could not understand. Table 3 and Figure 3 are also difficult to understand. Maybe you should delete them. Think about including in the main paper one of the Appendix Figures.
---

VERSION 1 – AUTHOR RESPONSE

Reviewer Reports:

Reviewer: 1

Prof. Antonio Pontiroli, University of Milan

Comments to the Author:

Insufficient weight loss five years after Roux-en-Y gastric bypass: Prevalence, metabolic consequences and prediction estimates -A prospective registry study

In a retrospective analysis of data prospectively collected from the Scandinavian Obesity Surgery Registry (SOReg), the study aimed to investigate the heterogeneity of weight loss five years after RYGB and the association with cardiometabolic health; in total, 23.1% met at least one definition of surgical treatment failure at five years, and surgical treatment failure at five years was predicted by combined demographic and anthropometric measures from baseline, one and two years post-surgery (area under the curve=0.874). The authors conclude that 23% met at least one criterion of surgical treatment failure, which was associated with a greater risk of relapse and a higher incidence of T2DM, dyslipidemia and hypertension five years after surgery. Poor initial weight loss and early weight regain are strong predictors of long-term treatment failure and may be used for early identification of patients who require additional weight loss support

The study is of interest. However, more details should be given for this retrospective study, whenever possible to retrieve.

1. For instance, even though the study is mainly centered on RYGB, we know from the SOS experience that many obese patients underwent other surgical procedures, such as VBG and LAGB. Do the authors have data on these two techniques? Are the SOS and the SOReg completely independent?

Response: Thank you for your thoughtful questions. We excluded SG, BPD, VBG and LAGB. The reason is that while SG has increased in usage over the last years, the participants in this study underwent surgery between 2007 to 2011 and during those years RYGB stood for 95-97.5% of all bariatric surgery done in Sweden (added on page 6 line 128). As can be noted from the study flowchart, only 46 patients underwent SG and 113 patients BPD and therefore excluded. We did not request data from patients undergoing VBG or LAGB as surgery volumes were very low (less than 100 cases per year).

We have revised the manuscript according to these concerns in the Methods section and clarified that the study is based on RYGB patients only (page 7 line 139). We have also added this to the strengths and limitation part of the discussion since the complete dominance of RYGB performed during these years reduces bias caused by possible differences in patient selection between procedures (page 17 line 378).

SOS and SOReg are completely independent. The SOS study was a controlled study which closed enrollment in the year 2000 and the national quality register "Scandinavian obesity surgery register" (SOReg) began collecting data in 2007.

2. An interesting point of the study is that patients unavailable were different from patients remaining in the follow-up study; do the authors have data on the psycho aspects of these patients, as the work up of patients undergoing bariatric surgery usually includes these aspects? Do they have data on the adherence of patients of the two groups to pre- and post surgery visits and exams, at least for the first two years after surgery?

Response: Thank you for raising this important question. The influence of psychological wellbeing is possibly an important factor for long-term outcome which deserves attention. We have under limitations (page 17 line 385) added that we unfortunately lack such information.

Regarding adherence, we have specified the percentage attending the clinic for the follow-ups of year one and two (eTable 1 Supplement) as well as a full description of modality at year five in the Methods (page 7 line 142).

3. Aside from classical metabolic risk factors, do the authors have data on sleep apnea and its profile over time?

Response: We did not study sleep apnea. Only very severe cases of sleep apnea, treated with CPAP are registered in SOREG which means that the number of patients is limited and the outcome problematic to evaluate as the only outcome possible to study is whether the patient is still on CPAP or not. However, we do acknowledge that it had been of value to study the association between sleep apnea and weight regain after surgery.

4. Do the authors have data on the predictive role of T2DM and of hypertension for weight loss in general and on the occurrence of STF?

Response: Thank you for pointing this out, which is a mistake on our part, this was supposed to be addressed in the discussion, and it has now been added, (page 16 line 351). Specific results can be found in supplementary appendix 2.

Minor points

In the text, it should be alluded to male sex and not to sex

Response: After carefully going through the manuscript, we have found one ambiguous phrasing and clarified that (page 14 line 309).

In the conclusions, the authors should address the unsolved question: is it that bariatric surgery does not respond to expectations in more severe cases? Do the authors have an overall different conclusion?

Response: This is indeed an important question but it goes beyond the topic of this particular study. It is not unlikely that the expectations of severe cases are rarely met as the total weight loss is poorly associated with the total fat mass. On the other hand our results do not indicate a strong association between starting BMI and surgical treatment failure although it was slightly more common in those with very high BMI and only 59 of 1371 patients with surgical treatment failure had no weight regain present (specified n=59 page 13 line 266). Appendix table 4, shows that the mean difference in starting BMI was 44.5 vs 42.5, a statistically significant finding but clinically less noticeable and well below BMI 50.

Reviewer: 2

Dr. Carlos Schiavon, Research Institute HCor

Comments to the Author:

Dear authors,

This is a very interesting paper about an important issue. Please, see my comments and try to respond them.

Title: I suggest to you t change the order of words: Prevalence of Insufficient weight loss five years after Roux-en-Y gastric bypass: metabolic consequences and prediction estimates - A prospective registry study

Objective: I suggest to you to include the prediction estimate in the objectives.

Thank you for your thoughtful questions and suggestions.

Title - We have rephrased the title according to your suggestion.

Objective – We agree that this should be mentioned in the abstract to more closely resemble the objectives described within the manuscript. Therefor we have added the following sentence to the abstract which now reads;

“Objective: The study aimed to investigate the heterogeneity of weight loss five years after RYGB and the association with cardiometabolic health as well as to model prediction estimates of surgical treatment failure.”

Strengths and limitations of this study: You should include in the limitation the number of patients (29%) who lost the FU because this data can interfere in the results because we know that many patients who do not return in the FU have obesity relapse.

Response: Thank you, we agree that lost to follow-up may underestimate the actual number of those with surgical treatment failure. We appreciate your input and have changed the sentence to be more precise (page 18 line 395)

Participants: Can you present the data (%) about each modality of clinical visit.

Response: Information about modality at the 5 year follow-up is presented in the flowchart, Figure 1. In addition we have specified the percentage attending the clinic for the follow-ups of year one and two (eTable 1 Supplement) as well as a full description of modality at year five in the Methods (page 7 line 142).

Page 1, line 13. You should change “incidence” by “de novo”

Response: Thank you for the suggestion, after careful consideration we find that incidence might remain better suited when describing the development of known diseases such T2D, dyslipidemia and hypertension. While de novo certainly could be used we believe it is more commonly used in context of genetic studies for newly occurring mutations. We are however willing to change the wording at the discretion of the editors.

Page 14: $P(\text{surgical treatment failure}) = \exp(a)/(1+(a))$ with $a = -1.1 + 0.00545*(\text{sexFemale}) + 0.00299*(\text{age at surgery}) + 0.14949*(\text{baseline BMI}) + 0.22310*(\%T\text{WL year one}) + 0.15982*(\text{weight change year one to year two (kg)})$.

You need to explain this formula. I really could not understand.

Table 3 and Figure 3 are also difficult to understand. Maybe you should delete them.

Response: We agree, the formula is not intuitively easy to catch. Within the appendix 3, we have tried to facilitate for the reader by giving examples of how this formula works. Essentially it presents the

coefficients from the prediction model which can be used e.g. in an excel spreadsheet to calculate the risk of surgical treatment failure in any given RYGB patient two years after surgery. We think it had been unfortunate to delete table 3 as it provides details on how this prediction is made. Similarly, figure 3 provides a graph showing how well the prediction model works and thereby provides a rational why it's usefulness.

Think about including in the main paper one of the Appendix Figures.

Response: Thank you for the suggestion. Although the figures in the appendix complement the findings presented in the study, they have to remain in the supplementary material as we consider the figure 3 and table 3 are more essential for the understanding of the core message of the study. .

Reviewer: 1

Competing interests of Reviewer: none

Reviewer: 2

Competing interests of Reviewer: None declared

VERSION 2 – REVIEW

REVIEWER	Pontiroli, Antonio University of Milan, medicina
REVIEW RETURNED	05-Feb-2021
GENERAL COMMENTS	no further concern
REVIEWER	CARLOS AURELIO SCHIAVON BP HOSPITAL, SAO PAULO BRAZIL
REVIEW RETURNED	16-Feb-2021
GENERAL COMMENTS	Dear authors, thanks for your corrections in the manuscript Title: Prevalence can be deleted after the ":" Please, correct this formula: %TWL was calculated as ((baseline BMI – year five BMI)/baseline BMI)*100 Were your patients operated from 2007 to 2011 or 2007 to 2012?